# KLUE: Korean Language Understanding Evaluation

**Sungjoon Park**[*1,4], **Jihyung Moon**[*1], **Sungdong Kim**[*2], **Won Ik Cho**[*8]
**Jiyoon Han**[†9], **Jangwon Park**, **Chisung Song**, **Junseong Kim**[6], **Youngsook Song**[11]
**Taehwan Oh**[†9], **Joohong Lee**[6], **Juhyun Oh**[†8], **Sungwon Lyu**[5], **Younghoon Jeong**[10]
**Inkwon Lee**[10], **Sangwoo Seo**[6], **Dongjun Lee**, **Hyunwoo Kim**[8], **Myeonghwa Lee**[2]
**Seongbo Jang**[6], **Seungwon Do**, **Sunkyoung Kim**[4], **Kyungtae Lim**[12], **Jongwon Lee**
**Kyumin Park**[4], **Jamin Shin**[7], **Seonghyun Kim**, **Lucy Park**[1]
**Alice Oh**[**4], **Jung-Woo Ha**[**2], **Kyunghyun Cho**[**3]

[1]Upstage, [2]NAVER AI Lab, [3]New York University, [4]KAIST,
[5]Kakao Enterprise, [6]Scatter Lab, [7]Riiid, [8]Seoul National University, [9]Yonsei University,
[10]Sogang University, [11]Kyung Hee University, [12]Hanbat National University

[*]sungjoon.park@kaist.ac.kr, [*]jihyung.moon@upstage.ai,
[*]sungdong.kim@navercorp.com, [*]tsatsuki@snu.ac.kr,
[**]alice.oh@kaist.edu, [**]jungwoo.ha@navercorp.com, [**]kyunghyun.cho@nyu.edu,

## Abstract

We introduce Korean Language Understanding Evaluation (KLUE) benchmark. KLUE is a collection of eight Korean natural language understanding (NLU) tasks, including Topic Classification, Semantic Textual Similarity, Natural Language Inference, Named Entity Recognition, Relation Extraction, Dependency Parsing, Machine Reading Comprehension, and Dialogue State Tracking. We create all of the datasets from scratch in a principled way. We design the tasks to have diverse formats and each task to be built upon various source corpora that respect copyrights. Also, we propose suitable evaluation metrics and organize annotation protocols in a way to ensure quality. To prevent ethical risks in KLUE, we proactively remove examples reflecting social biases, containing toxic content or personally identifiable information (PII). Along with the benchmark datasets, we release pretrained language models (PLM) for Korean, KLUE-BERT and KLUE-RoBERTa, and find KLUE-RoBERTa$_{\text{LARGE}}$ outperforms other baselines including multilingual PLMs and existing open-source Korean PLMs. The fine-tuning recipes are publicly open for anyone to reproduce our baseline result. We believe our work will facilitate future research on cross-lingual as well as Korean language models and the creation of similar resources for other languages. KLUE is available at `https://klue-benchmark.com/`.

## 1   Introduction

A major factor behind the recent success of pretrained language models, such as BERT [30] and its variants [84, 22, 49] as well as GPT-3 [112] and its variants [113, 78, 9], has been the availability of well-designed benchmark suites for evaluating their effectiveness in natural language understanding (NLU). GLUE [135] and SuperGLUE [134] are representative examples of such suites and were

---

[*]Equal Contribution. A description of each author's contribution is available at the end of paper.
[**]Corresponding Authors.
[†]Work done at Upstage.

35th Conference on Neural Information Processing Systems (NeurIPS 2021) Track on Datasets and Benchmarks.

designed to evaluate diverse aspects of NLU, including syntax, semantics and pragmatics. The research community has embraced GLUE and SuperGLUE and has made rapid progress in developing better model architectures as well as learning algorithms for NLU.

The success of GLUE and SuperGLUE has sparked interests in building standardized benchmark suites for other languages. Such efforts have been pursued along two directions. First, various groups in the world have independently created language-specific benchmark suites; a Chinese version of GLUE (CLUE [144]), a French version of GLUE (FLUE [74]), an Indonesian variant [139], an Indic version [57] and a Russian variant of SuperGLUE [127]. On the other hand, some have relied on both machine and human translation of existing benchmark suites for building multilingual versions of the benchmark suites that were created initially in English. These include XGLUE [80] and XTREME [54]. Although the latter approach scales much better than the former, the latter often fails to capture the sociocultural aspects of NLU and also introduces various artifacts arising from translation.

Hence, we build a new benchmark suite, Korean Language Understanding Evaluation (KLUE), for Korean which is the 13-th most used language in the world according to [34] but lacks a unified benchmark suite for NLU. Instead of starting from existing benchmark tasks or corpora, we build this benchmark suite from ground up by determining and collecting the base corpora, identifying a set of benchmark tasks, designing appropriate annotation protocols, and finally validating the collected annotations. This allows us to preemptively address and avoid properties that may have undesirable consequences, such as copyright infringement, annotation artifacts, social biases and privacy violations.

In summary, our contributions are:

- We build a new benchmark suite, Korean Language Understanding Evaluation (KLUE), consisting of eight NLU tasks constructed from scratch in a principled way

- We build and publicly release pretrained language models for Korean with introducing Korean-aware tokenization method.

## 2   KLUE Benchmark

### 2.1   Design Principles

We design KLUE with the following principles:

- *Covering diverse tasks and corpora*: To cover diverse aspects of language understanding, we choose eight tasks that address various task formats and domains of source corpora, including news, encyclopedia, user reviews, smart home queries and task-oriented dialogue. As Korean has distinct styles of formal and colloquial language, we explicitly include both.

- *Accessible to everyone without any restriction*: It is important for a benchmark suite to be available to everyone, such that the benchmark serves as a standard for evaluating and improving NLU systems. We thus ensure any corpora and resources in KLUE can be freely copied, redistributed, remixed and transformed for the purpose of benchmarking NLU systems.

- *Obtaining accurate and unambiguous annotations*: Ambiguity in benchmark tasks leads to less reliable evaluation, which often results in the discrepancy between the quality of an NLU system measured by the benchmark and its true quality. In order to minimize such discrepancy, we carefully design annotation guidelines of all tasks and improve them over multiple iterations, to assure accurate annotations.

- *Mitigating ethical issues in PLMs*: It has been repeatedly observed that large-scale language models often amplify the social biases in the training data [97]. We proactively remove examples from both unlabeled and labeled corpora that reflect social biases, contain toxic content or personally identifiable information (PII), both manually and automatically. Social biases are defined as overgeneralized judgment on certain individuals or groups based on social attributes (e.g., gender, ethnicity, religion). Toxic contents include insults, sexual harassment, and offensive expressions.

Table 1: Source corpora chosen for building KLUE. The top section, *News Headlines* is not protected by copyright act since they are not classified as a work due to their lack of creativity. The middle section is a collection of corpora under the permissive licenses. The bottom section, KED and Acrofan, is originally prohibited from creating derivative works, however, we release such condition by exclusive contract. For the column, *Volume*, we denote *Small* as corpus size under 1k, *Medium* as in between 1k and 50k, and *Large* as over 50k. All source corpora we collected, selection processes, and details on selected corpora are described in Appendix B.

| Dataset | License | Domain | Style | Ethical Risks | Volume | Contemporary Korean |
|---|---|---|---|---|---|---|
| News Headlines | N/A | News (Headline) | Formal | Low | Large | o |
| Wikipedia | CC BY-SA 3.0 | Wikipedia | Formal | Low | Large | o |
| Wikinews | CC BY 2.5 | News | Formal | Low | Small | o |
| Wikitree | CC BY-SA 2.0 | News | Formal | Medium | Large | o |
| Policy News | KOGL Type 1 | News | Formal | Low | Medium | o |
| ParaKQC | CC BY-SA 4.0 | Smart Home Utterances | Colloquial | Low | Medium | o |
| Airbnb Reviews | CC0 1.0 | Review | Colloquial | Medium | Large | o |
| NAVER Sentiment Movie Corpus (NSMC) | CC0 1.0 | Review | Colloquial | Medium | Large | o |
| Acrofan News | CC BY-SA 4.0 for KLUE-MRC by Contract | News | Formal | Low | Large | o |
| The Korea Economics Daily News | CC BY-SA 4.0 for KLUE-MRC by Contract | News | Formal | Low | Large | o |

## 2.2 Source Corpora

We have actively sought corpora that are accessible, cover diverse domains and topics, and are written in modern Korean. This active search has resulted in ten sources from which we derive task-specific corpora in Table 1. These base corpora are released under CC BY(-SA) license or not considered as copyrighted work, permitting 1) derivative work, 2) redistribution, and 3) commercial use. Then we carefully preprocess them because the collected corpora came from various sources with varying levels of quality and curation. We remove noise, toxic or socially biased content, and PII, using predefined rules and machine learning models.

## 2.3 Considerations in Annotation

For all tasks in KLUE, we annotate a subset from the source corpora. We take into account three major considerations below:

- *Better reflection of linguistic characteristics of Korean*: Many existing Korean datasets were constructed as a part of multilingually aligned benchmarks, and they do not fully reflect linguistic characteristics of Korean such as agglutinative nature in named entity recognition (NER) [102], or tagset in part-of-speech (POS) tagging and dependency parsing (DP) [88, 46]. We write and revise annotation guidelines more appropriate for the linguistic properties of Korean.

- *Obtaining accurate annotations*: We provide crowdworkers or selected participants with a carefully designed annotation guideline and improve it over multiple iterations, in order to reduce the ambiguity in the annotation process as well as to mitigate the known artifact issues. In particular, we often filter out examples for which annotators cannot easily agree on.

- *Mitigating harmful social bias and removing PII*: To not incentivize socially biased NLU systems [7], we explicitly instruct both annotators and inspectors to manually mark and/or exclude examples that are unacceptable according to our principle of ethics. Our definitions of *bias* and *hate speech* follow Moon et al. [94]. We denote *bias* as an overgeneralized prejudice on certain groups or individuals based on the following traits: gender, race, background, nationality, ethnic group, political stance, skin color, religion, disability, age, appearance, (socio-)economic status, and occupations. In the case of *hate speech*, we include offensive, aggressive, insulting, or sarcastic contents. To deal with privacy risks, we identify a list of personally identifiable information (PII)

Table 2: Task Overview

| Name | Type | Format | Eval. Metric | # Class | {\|Train\|, \|Dev\|, \|Test\|} | Source | Style |
|------|------|--------|--------------|---------|----------------------------------|--------|-------|
| KLUE-TC (YNAT) | Topic Classification | Single Sentence Classification | Macro F1 | 7 | 45k, 9k, 9k | News (Headline) | Formal |
| KLUE-STS | Semantic Textual Similarity | Sentence Pair Regression | Pearson's $r$, F1 | [0, 5] 2 | 11k, 0.5k, 1k | News, Review, Query | Colloquial, Formal |
| KLUE-NLI | Natural Language Inference | Sentence Pair Classification | Accuracy | 3 | 25k, 3k, 3k | News, Wikipedia, Review | Colloquial, Formal |
| KLUE-NER | Named Entity Recognition | Sequence Tagging | Entity-level Macro F1 Character-level Macro F1 | 6, 13 | 21k, 5k, 5k | News, Review | Colloquial, Formal |
| KLUE-RE | Relation Extraction | Single Sentence Classification (+2 Entity Spans) | Micro F1 (without *no_relation*), AUPRC | 30 | 32k, 8k, 8k | Wikipedia, News | Formal |
| KLUE-DP | Dependency Parsing | Sequence Tagging (+ POS Tags) | Unlabeled Attachment Score, Labeled Attachment Score | # Words, 38 | 10k, 2k, 2.5k | News, Review | Colloquial, Formal |
| KLUE-MRC | Machine Reading Comprehension | Span Prediction | Exact Match, ROUGE-W (LCCS-based F1) | 2 | 12k, 8k, 9k | Wikipedia, News | Formal |
| KLUE-DST (WoS) | Dialogue State Tracking | Slot-Value Prediction | Joint Goal Accuracy Slot Micro F1 | (45) | 8k, 1k, 1k | Task Oriented Dialogue | Colloquial |

following KISA (Korea Internet and Security Agency) guideline,[1] whose information is related to a living individual based on personal information protection act of Korea.[2] We do not consider public figure's name as personal information.[3]

## 2.4 Tasks

We carefully choose the following eight tasks to cover diverse aspects of NLU in Korean while minimizing redundancy among the tasks. In Table 2, we illustrate important properties of the tasks, such as type, format, evaluation metrics, and annotated data characteristics.

Here, we list the tasks and describe why we include the task in KLUE, how we manage the construction process, and how/why we choose the evaluation metrics. For all tasks, we guide annotators to report examples that contain hate speech, biased expressions, or PII, to remove them from our benchmark[4].

**KLUE-TC**  Topic classification (TC) is a single sentence classification task to predict the topic of a given text snippet. We include TC in our KLUE benchmark, as inferring the topic of a text is a key capability that should be possessed by a language understanding system. As a typical single sentence classification task, other NLU benchmarks such as CLUE [144] and IndicGLUE [57] also contain TNEWS and News Category Classification. For Korean, no dataset has been proposed for the task, which motivates us to construct the first Korean topic classification benchmark.

In KLUE-TC, given a news headline, a text classifier must predict a topic which is one of {politics, economy, society, culture, world, IT/science, sports}. We formulate TC as a single sentence classification task following previous works. The evaluation metric of KLUE-TC is a macro-F1 score to give the same importance to each class.

---

[1]https://www.kisa.or.kr/public/laws/laws2_View.jsp?cPage=1&mode=view&p_No=282&b_No=282&d_No=3

[2]https://www.law.go.kr/LSW//lsInfoP.do?lsiSeq=213857&chrClsCd=010203&urlMode=engLsInfoR&viewCls=engLsInfoR#0000

[3]See the precedent set by the Supreme Court in Korea: 대법원 2011. 9. 2. 선고 2008다42430 전원합의체 판결 available at https://glaw.scourt.go.kr/wsjo/panre/sjo100.do?contId=2060159&q=2008%EB%8B%A442430.

[4]All the workers were guaranteed to be paid minimum wage in Korea (about $7.5 per hour).

We collect news headlines from online articles distributed by Yonhap News Agency (YNA) and manually annotate the topics of the headlines, which is to address the gap between the headline and the predefined category. 13 selected workers labeled topics for 70,000 headlines. For each headline, 3 workers annotated topics. We filter invalid headlines and keep headlines whose topic is agreed by at least two annotators out of three, leaving 63,892 examples. For more information, see Appendix C.

**KLUE-STS**    Semantic textual similarity (STS) is a regression task to measure the degree of semantic equivalence between two sentences. We include STS in our benchmark because it is essential to other NLP tasks such as machine translation, summarization, and question answering. Like STS [13] in GLUE [135], many NLU benchmarks include comparing semantic similarity of text snippets such as semantic similarity [144], paraphrase detection [135, 57], or word sense disambiguation [127, 74].

We formulate STS as a sentence pair regression task which predicts the semantic similarity of two input sentences as a real value. A model performance is measured by Pearson's correlation coefficient following the evaluation scheme of STS-b [13]. We additionally binarize the real numbers into two classes (paraphrased or not) with a threshold score, and use F1 score to evaluate the model.

We carefully sample and generate sentence pairs to cover all range of the similarities. For unlabeled corpora AIRBNB (colloquial review), POLICY (formal news), round-trip translation (RTT) is used to generate similar sentence pairs and greedy sentence matching (GSM) to sample less similar pairs. For labeled dataset, PARAKQC [18] (smart home utterances), we leverage the labeled intents of the commands to pair both similar and dissimilar sentences. 19 workers are employed and annotated the similarity between two sentences in integers from 0 (no meaning overlap) to 5 (meaning equivalence). We remove outlier annotations and then take average from the remaining labels for the final labels. The total number of KLUE-STS is 13,224 sentence pairs. Details are described in Appendix D.

**KLUE-NLI**    The goal of natural language inference (NLI) is to train a model to infer the relationship between the *hypothesis* sentence and the *premise* sentence. Given a *premise*, an NLI model determines if *hypothesis* is true (entailment), false (contradiction), or undetermined (neutral). The task is also known as recognizing textual entailment (RTE) [27]. NLI datasets are also included in several NLU benchmarks such as GLUE [135] and superGLUE [134], and they are valuable as training data for other NLU tasks [24, 109, 117], which leads us to include NLI task in KLUE.

We formulate NLI as a classification task where an NLI model reads each pair of *premise* and *hypothesis* sentences and predicts whether the relationship is entailment, contradiction, or neutral. We use the classification accuracy to measure the model performance.

We construct KLUE-NLI by using a collection method similar to that of SNLI [8] and MNLI [140], while avoiding known annotation artifacts. We select premise sentences from multi-source corpora which allows to generate hypotheses. Then for each premise sentence, we ask one annotator to generate three hypothesis sentences that correspond to the three relationship classes, each. The writer is trained enough to aware of the annotation artifacts. To validate the generated pairs, we ask four additional annotators to label the relationship. The final output label is the majority of the five annotations - one original label and four additional ones. KLUE-NLI consists of 30,998 sentence pairs. In Appendix E, we compare ours against SNLI and MNLI dataset, and provide the details of the construction process and analysis.

**KLUE-NER**    Named entity recognition (NER) is a sequence tagging task to detect the boundaries of named entities in unstructured text and identify the entity type. An entity can be a sequence of words that refers to a person, location, organization, time expression, quantity, or monetary value. Since NER is important for application fields like syntactic analysis, goal-oriented dialog systems, question and answering, and information extraction, many NLU benchmarks contain NER datasets [139, 57, 80, 54]. Despite the rise of necessity of NER datasets of various domains and styles, there are few existing Korean NER datasets to cover such needs.

In KLUE-NER, a model should detect the spans and classify the types of entities included in an input sentence. The six entity types used in KLUE-NER are person, location, organization, date, time, and quantity. We tag entity types with character-level BIO (Begin-Inside-Outside) scheme, as Korean word is mostly a combination of named entity and particle. To respect such characteristics, we evaluate a model performance using traditional entity-level F1 score and newly proposed character-level F1 score.

We choose WIKITREE (news articles) and NSMC (reviews) as the source corpora and sample sentences from them to mix the styles of written and spoken languages. Our six tag sets adapt Korean tag sets defined in Korean Telecommunications Technology Association (TTA) NER guidelines and colloquial tag sets defined MUC-7 [16]. 51 crowdworkers did the annotation first, and then two linguists validate the results. To correct erroneous annotations even after validation, six NLP researchers manually correct the annotation errors, resulting 31,008 sentences. Annotation scheme and statistics are attached in Appendix F.

**KLUE-RE** Relation extraction (RE) is a task to identify semantic relations between entity pairs. The relation is defined between an entity pair consisting of *subject entity* ($e_{subj}$) and *object entity* ($e_{obj}$). For example, in a sentence 'Kierkegaard was born to an affluent family in Copenhagen', the subject entity is 'Kierkegaard' and the object entity is 'Copenhagen'. The goal is then to pick an appropriate relationship between these two entities; '*place_of_birth*'. To ensure KLUE-RE captures this aspect of language understanding, we include a large-scale RE benchmark. Because there is no large-scale RE benchmark publicly available in Korean, we collect and annotate our own dataset.

We formulate RE as a single sentence classification task. A model picks one of predefined relation classes describing the relation between two entities within a given sentence. In other words, an RE model predicts an appropriate relation $r$ of entity pair ($e_{subj}$, $e_{obj}$) in a sentence $s$, where $e_{subj}$ is the subject entity and $e_{obj}$ is the object entity. We refer to ($e_{subj}$, $r$, $e_{obj}$) as a relation triplet. The entities are marked as corresponding spans in each sentence $s$. There are 30 relation classes that consist of 18 person-related relations, 11 organization-related relations, and *no_relation*. Detailed explanation of these classes are presented in Table 13. We evaluate a model using micro F1 score, computed after excluding *no_relation*, and area under the precision-recall curve (AUPRC) including all 30 classes.

We draw sentences from WIKIPEDIA and news articles (WIKITREE and POLICY) to cover various sets of named entities and relational facts. Then we apply existing NER models to leave examples having at least two named entities, marking $e_{subj}$ and $e_{obj}$, and sample sentences from those in two distinct ways. First is random sampling, which is similar to a real world scenario where the pair is highly likely to be irrelevant (*no_relation*). Second is distant supervision, which leverages Korean KB to have more chance to include relation-existing pairs. Our 30 relation classes are based on Text Analysis Conference Knowledge Base Population (TAC-KBP) [89]. We employ 163 qualified workers and assign three workers to each sentence to label the relation, taking majority-vote labels as gold labels. KLUE-RE consists of 48,001 annotated sentences. More details are in Appendix G.

**KLUE-DP** Dependency parsing (DP) aims to find the relations among words in a sentence. It is an important component in many NLP systems because it captures the syntactic structure of a sentence. We include DP in KLUE to evaluate the representational power of language models in terms of syntactic features.

Formally, a dependency parser predicts a graph structure of an input sentence based on the dependency grammar [29, 28]. In general, a parse tree consists of dependency arcs, connecting dependents to their heads, and the dependency labels attached to the arcs that represent the relations between dependents (DEPREL) and their heads (HEAD). Since each word in a sentence has a pair of dependency information (HEAD, DEPREL), we formulate DP as a word-level sequence tagging task. We evaluate a model's performance using unlabeled attachment score (UAS) and labeled attachment score (LAS).

For the source, we sample sentences from both WIKITREE (formal) and AIRBNB (informal). We annotate part-of-speech on the corpus in advance and use them when annotating the dependency relations. Both POS and DP are annotated and cross-validated by ten Korean PhD students who are majoring in linguistics. The final KLUE-DP consists of 14,500 sentences. The comprehensive process is demonstrated in Appendix H with dataset statistics.

**KLUE-MRC** Machine reading comprehension (MRC) is a task designed to evaluate a model's ability to read a given passage and then answer a question about the passage. Most existing MRC benchmarks are in English [21, 56, 60, 114, 115, 147, 152], and they are widely used in evaluating pre-trained language models for text comprehension. In Korean, however, an appropriate MRC benchmark is not available because existing Korean MRC datasets are less challenging, limited in access, or simply machine-translated from an English dataset [81, 1, 76]. We therefore include MRC in KLUE and create a new challenging Korean MRC benchmark. When building KLUE-MRC, we

consider providing multiple question types, preventing reasoning shortcuts when answering to a multi-hop question, and using passage from several domains without any copyright violation.

We formulate MRC as a task of predicting an answer span of a question from a given text passage. A model input is a concatenated sequence of a question and a passage separated with a delimiter. A model output is start and end positions of a predicted answer span within a passage. If the question is unanswerable within the given passage, the model should predict the empty answer string. We evaluate a model with two metrics: 1) exact match (EM) and 2) character-level ROUGE-W. Note that character-level ROUGE-W is newly proposed character-level metric instead of character-level F1 score which have commonly used in other Korean MRC datasets. We find character-level F1 score could overestimate a model's performance as it gives score to any character overlap regardless of a sequential order.

We collect passages from Korean WIKIPEDIA and news articles provided by The Korea Economy Daily and ACROFAN. On each paragraph, annotators create questions 1) by paraphrasing a sentence in the passage, 2) requiring multi-sentence reasoning, and 3) that are unanswerable. Answers are annotated at the same time. To make question and answer pairs without having known artifacts, we prepare the guideline with specific do's and don'ts and train employed workers thoroughly. KLUE-MRC consists of 12,207 paraphrasing-based questions, 7,895 multi-sentence reasoning questions, and 9,211 unanswerable questions, for a total of 29,313 from 22,343 documents and 23,717 passages. We provide further information on each question type, statistics, and in-depth analyses in Appendix I.

**KLUE-DST**    Dialogue State Tracking (DST) is about predicting *dialogue states* from a given task-oriented dialogue. Several recent papers have considered task-oriented dialogue (TOD) as an important problem of natural language understanding. For instance, DecaNLP [87] includes a DST, which is a key component of TOD, into one of their benchmark tasks, while DialoGLUE [90] releases the first task-oriented dialogue benchmark containing various sub-tasks including DST.

Specifically, DST is a task to predict slot (e.g. hotel type) and value (e.g. guest house, hotel, motel) pairs after each user utterance. The potential pairs are predefined by a task schema and knowledge base (KB), tied to the choice of a scenario. For evaluation, we use joint goal accuracy (JGA) and slot micro F1 score. JGA checks if all of the predicted slot-value pairs are exactly matched with the ground-truth for every turn, while the slot micro F1 computes F1 score for each slot-value pair independently. We also name this task as Wizard of Seoul (WoS).

We define a task schema and create a knowledge base, and then design an annotation system based on the schema. Then, we collect task-oriented dialogues with dialogue state annotations by following 'Self-dialog' scheme which requests a single worker to play both user and system roles [11]. Crowdworkers generate dialogues and the corresponding states by using the system. WoS contains overall 10,000 dialogues with 146,692 turns across 5 domains. Further information on the process is provided in Appendix J.

# 3   Experiments

In order to facilitate further research using KLUE, we provide strong baselines for all the benchmark tasks within it. As a part of this effort, we pretrain and release large-scale language models for Korean, which will reduce the burden of retraining these models from individual researchers. We also compare our models with existing multilingual pretrained language models and open-sourced Korean-specific models on the proposed KLUE benchmark.

## 3.1   Pretrained Language Models

**Pretraining Corpora**    We collect publicly available data from diverse sources to cover a broad set of topics and many different styles. Having noticed quite a bit of PII and undesirable social biases in these large corpora, we pseudonymize PII while do not filter out socially biased contents nor hate speech for three reasons. First, manual inspection is infeasible. Second, it is a challenging problem on its own to automatically detect socially biased contents or hate space. Lastly, being blind to such harmful contents prevents the future use of a language model for detecting and correcting these harmful contents. Details of our choices are described in Appendix K.1.

**Pretraining Korean Language Models** We pretrain language models, namely KLUE-BERT and KLUE-RoBERTa, following the similar recipes of BERT [30] and RoBERTa [84], respectively. The models are trained on sequences of at most 512 tokens long with a static or dynamic masking strategy following the original training procedure. We use whole word masking (WWM) which masks all of the tokens that form a single word. We set the batch size to 256 for BERT and 2048 for RoBERTa and fix the learning rate to $10^{-4}$ for both. For the pretraining corpus, we gather publicly available Korean corpora of size approximately 62GB from diverse sources to cover a broad set of topics and many different styles. The most distinct part is tokenization. We use morpheme-based subword tokenization instead of BPE, considering the aggulutinative nature of Korean. The details of our pretraining are in Appendix K.

**Comparison Models** In addition to our own language models, we evaluate two existing multilingual language models and two Korean monolingual language models on our benchmark. For multilingual models, we employ mBERT [30] and XLM-R [26]. For the Korean models, we compare KR-BERT [77] and KoELECTRA [104]. You can find further information on each model in Appendix K.

## 3.2 Fine-tuning Language Models

**Single Sentence Classification** For **KLUE-TC**, we follow single sentence classification architecture in [30]. **KLUE-RE** on the other hand requires a special procedure to indicate entities within the input sentence. We use `<subj>`, `</subj>`, `<obj>`, and `</obj>` to mark the beginnings and the ends of subject and object entities, respectively, following Baldini Soares et al. [5].

**Sentence Pair Classification and Regression** For **KLUE-NLI**, a sentence pair classification task, we adopt the same approach to sentence pair classification framework suggested by Devlin et al. [30]. While for **KLUE-STS**, only the final layer is different, as **KLUE-STS** is a a regression task.

**Multiple-Sentence Slot-Value Prediction** **WoS** is a slot-value prediction task for a given dialogue context, where the prediction should be considered across multiple turns instead of a single utterance. We employ an encoder-decoder model following the architecture of TRADE [142], which consists of an utterance encoder, a state generator, and a slot gate classifier. In our implementation, we change the utterance encoder from GRU [17] to pretrained language model to get better representations. We also modify the slot gate classifier to predict additional two slot gate labels (*yes*, *no*), since WoS contains relatively more Boolean type slots than MultiWOZ [10]. We jointly minimize the cross-entropy loss of the state generator and slot gate classifier.

**Sequence Tagging** **KLUE-NER** is a subword-level tagging task and **KLUE-MRC** is a span prediction task, and in both tasks, each token is linearly mapped to a predefined label. We employ the same architecture provided in [30] and only use the given dataset for finetuning. We frame **KLUE-DP** as a sequence tagging problem. Our baseline architecture follows the model proposed in [39]. In our implementation, we use a pretrained language model to extract subword representations and concatenate the first and last subword token representations of each word, to form word vector representations, since the annotation is done at the word level. For the attention layers, we use biaffine attention [33] to predict the head, and bilinear attention [66] to predict the arc type for each word. Cross-entropy loss is minimized to tune all the parameters.

**Results** We present all results in Table 3 and summarize few observations. First, Korean monolingual models generally outperform multilingual models. Second, for sequence tagging tasks like KLUE-NER and KLUE-MRC, the tokenization matters. Although XLM-R$_{\text{LARGE}}$ is on par with KLUE-BERT$_{\text{BASE}}$ on KLUE-MRC based on character-level score ROUGE, the performance gap is wider on entity-level score EM. Similar results are shown in KLUE-NER, and this demonstrates the effect of out tokenization method. Third, different models perform best on different tasks when controlled for their sizes; KLUE-BERT performs best for YNAT, KLUE-RoBERTa for KLUE-RE, KLUE-DP, KLUE-MRC and WoS, and KoELECTRA$_{\text{BASE}}$ for KLUE-STS, KLUE-NLI, and KLUE-NER. Lastly, as we increase the model size, KLUE-RoBERTa$_{\text{LARGE}}$ outperforms the other models in all tasks except for YNAT and KLUE-NER. More details of our experiments and further analyses are in Appendix L.

Table 3: Evaluation results of our pretrained LMs and other baselines on KLUE benchmark test set. The F1 refers to a macro-F1 score. The $F1^E$ and $F1^C$ of KLUE-NER indicates entity-level and character-level macro-F1 score, respectively. The $F1^{mic}$ of KLUE-RE is micro-averaged F1 score ignoring the *no_relation*. The $F1^S$ of WoS is an average of slot-value pair level micro-F1 scores. The $R^P$ of KLUE-STS denotes Pearson correlation. **Bold** shows the best performance across the models, and underline indicates the best performance among BASE models.

| Model | YNAT | KLUE-STS | | KLUE-NLI | KLUE-NER | | KLUE-RE | | KLUE-DP | | KLUE-MRC | | WoS | |
|---|---|---|---|---|---|---|---|---|---|---|---|---|---|---|
| | F1 | $R^P$ | F1 | ACC | $F1^E$ | $F1^C$ | $F1^{mic}$ | AUC | UAS | LAS | EM | ROUGE | JGA | $F1^S$ |
| mBERT$_{BASE}$ | 81.55 | 84.66 | 76.00 | 73.20 | 76.50 | 89.23 | 57.88 | 53.82 | 90.30 | 86.66 | 44.66 | 55.92 | 35.46 | 88.63 |
| XLM-R$_{BASE}$ | 83.52 | 89.16 | 82.01 | 77.33 | 80.37 | 92.12 | 57.46 | 54.98 | 89.20 | 87.69 | 27.48 | 53.93 | 39.82 | 89.61 |
| XLM-R$_{LARGE}$ | **86.06** | 92.97 | 85.86 | 85.93 | 82.27 | **93.22** | 58.39 | 61.15 | 92.71 | **88.70** | 35.99 | 66.77 | 41.20 | 89.80 |
| KR-BERT$_{BASE}$ | 84.58 | 88.61 | 81.07 | 77.17 | 74.58 | 90.13 | 62.74 | 60.94 | 89.92 | 87.48 | 48.28 | 58.54 | 45.33 | 90.70 |
| KoELECTRA$_{BASE}$ | 84.59 | 92.46 | 84.84 | 85.63 | 86.11 | 92.56 | 62.85 | 58.94 | 92.90 | 87.77 | 59.82 | 66.05 | 41.58 | 89.60 |
| KLUE-BERT$_{BASE}$ | 85.73 | 90.85 | 82.84 | 81.63 | 83.97 | 91.39 | 66.44 | 66.17 | 89.96 | 88.05 | 62.32 | 68.51 | 46.64 | 91.61 |
| KLUE-RoBERTa$_{SMALL}$ | 84.98 | 91.54 | 85.16 | 79.33 | 83.65 | 91.14 | 60.89 | 58.96 | 90.04 | 88.14 | 57.32 | 62.70 | 46.62 | 91.44 |
| KLUE-RoBERTa$_{BASE}$ | 85.07 | 92.50 | 85.40 | 84.83 | 84.60 | 91.44 | 67.65 | 68.55 | 93.04 | 88.32 | 68.67 | 73.98 | 47.49 | 91.64 |
| KLUE-RoBERTa$_{LARGE}$ | 85.69 | **93.35** | **86.63** | **89.17** | 85.00 | 91.86 | **71.13** | **72.98** | **93.48** | 88.36 | **75.58** | **80.59** | **50.22** | **92.23** |

# 4 Discussion

We develop KLUE with the aim of facilitating Korean NLP research, in response to the recent active development efforts of large Korean language models [63]. The entire NLP community has seen BERT [30] and its variants outperforming the previous NLU models for GLUE [135] and SuperGLUE [134], as well as the more recent GPT3 [9] with outstanding performance without fine-tuning (and with *in-context learning*) in natural language understanding and generation. Motivated by these models, many Korean researchers at various institutions rushed to pretrain large-scale Transformer-based Korean language models. Consequently, a number of nearly identical pretrained language models have been released to open-source communities. However, we could not systematically understand the behaviors and characteristics of these models because of the lack of well-designed general-purpose benchmarks like GLUE for Korean. KLUE will allow researchers to conduct controlled experiments to understand how and why various Korean LMs perform on certain tasks and thus obtain detailed insights into those models. Furthermore, since KLUE includes many representative NLU tasks that are also conducted in other languages, KLUE will function as a fundamental resource to NLP researchers who aim to conduct multilingual research with Korean and other languages.

**From Scratch vs. Translation** Translation has been the most straightforward approach for expanding English dataset to other languages. However, we insist to build KLUE from scratch to assure quality, considering its impact and role as the first Korean benchmark. To examine the quality difference quantitatively, we compare the correctness of sentence pairs and the corresponding labels of KLUE-NLI test set against that of KorNLI [45], a post-edited XNLI [25]. The results are shown in Table 4.

Table 4: Statistics of re-annotation results on randomly sampled 100 sentences from KorNLI and KLUE-NLI. Annotators are four native Korean undergraduates who are majoring in Korean linguistics and did not participate in the KLUE-NLI construction process. We compute the agreements of the additional labels to the gold labels for each dataset.

| Statistics (n = 100) | KorNLI | KLUE-NLI |
|---|---|---|
| Unanimous Gold Label (4 Agree) | 38.00% | **71.00%** |
| 3 Agree with Gold Label | 18.00% | 24.00% |
| 2 Agree with Gold Label | 18.00% | 3.00% |
| 1 Agrees with Gold Label | 16.00% | 2.00% |
| 0 Agrees with Gold Label | 10.00% | 0.00% |
| Individual Label = Gold Label | 64.50% | **91.00%** |
| No Gold Label (No 3 Labels Match) | 4.00% | **0.00%** |
| Majority Vote $\neq$ Gold Label | 26.00% | **0.00%** |

These numbers suggest that even human translation does not guarantee the quality. For KorNLI, annotators often report that they do not quite understand at least one of the two sentences or choose NEUTRAL because it is difficult to distinguish the semantic relationships of the sentences. On

the other hand, for KLUE-NLI, there is no case where the annotators struggle to grasp the logical semantic relationship. Given that 83% of French XNLI recovers the original English consesus label [25], KorNLI seems to lost relatively more semantic meanings/relationships during the translation. This results imply that the difference in characteristics between the source and target language might have affected.

**Overall Evaluation**    We do not average all scores gained from each task in KLUE. The performance of all tasks are measured by different evaluation metrics. This is because we carefully choose the metric for each task with considering its own characteristics. Their granularity differs by tasks, for example, KLUE-MRC and KLUE-NER employ character-level metrics because an entity can exist within a word in Korean whereas KLUE-STS and KLUE-NLI use sentence-level metrics. Furthermore, we use various metrics across tasks, such as F1 score, accuracy, AUPRC, UAS, LAS, ROUGE-W, joint goal accuracy, and Pearson's correlation. In this situation, simply computing the average of all tasks as in GLUE [135] results in misleading overall performance measure. The average will lose its interpretability as well as giving higher weights to a certain task in unintended ways. Accordingly, an alternative way to estimate a model's NLU capability is necessary. Recently, analyzing correctness of a model's prediction by using Item Response Theory (IRT) framework to estimate such capability is proposed [72], however, we find that it is not clear how it should be applied precisely in our benchmark. As of now, we thus decide to evaluate a model for each task separately without summarizing overall performance measure. This is our limitation, and we leave this problem for the future.

**Ethical Consideration**    To secure and maximize the continued availability and usefulness of the benchmark, we include only source corpora for which we know we can release under a license that permits both redistribution and re-mix without any restriction on the use. Furthermore, we first automatically detect hate speech and gender-biased sentences using toxicity classifiers and remove those. Annotators are clearly instructed to mark any instance that exhibits social biases and/or is toxic. Finally, we manually examine these marked sentences to exclude them from the final dataset. For PII, we rely on manual inspection during annotation. We discard any sentences that was reported to contain PII after manual inspection. See Appendix M for our full statement of ethical considerations.

**Broader Impact**    We distribute KLUE under CC BY-SA. The license allows everyone to freely copy and redistribute our benchmarks in any medium or format. In addition, one can improve our benchmark to build more challenging datasets after performance saturation. To function as a NLU *benchmark*, open access is a must. To set a good precedent for open access of data, we allow using our datasets for 1) any purpose, 2) derivative work, and 3) redistribution, as long as the existing copyrights in our benchmark datasets are respected. We also open our pretrained Korean language models and the implementation of pretraining and fine-tuning pipelines. This enhances reproducibility of our work, and allows anyone to fix and improve our data and models. We hope to contribute to the Korean NLP research community as well the wider NLP community.

## 5    Conclusion

We present KLUE, a suite of Korean NLU benchmarks that includes eight diverse tasks. KLUE is available for everyone, along with Korean language models trained to outperform multilingual models and other existing open-sourced Korean language models. We set high standards from the outset, as building the benchmark and training the models from scratch. We designed the benchmark datasets and trained the annotators rigorously to consider potential ethical issues including private information and hate speech. We documented in detail all of the benchmark construction and testing processes. We also discussed broader impacts and limitations of KLUE and our models. Despite the limitations, KLUE and the accompanying language models will facilitate future Korean NLP research by setting a valuable precedent describing how datasets and language models should be created and spread to a wider community.

## Acknowledgments and Disclosure of Funding

Data annotation costs were provided by Upstage, NAVER CLOVA, Scatter Lab, SelectStar, Riiid!, DeepNatural and KAIST. The leaderboard is built and supported by Upstage. GPU cloud computing is provided by NAVER CLOVA NSML [64], Google TensorFlow Research Cloud (TFRC), and Kakao Enterprise BrainCloud. These three computing resources were used to pretrain and fine-tune the language models. News articles for the MRC datasets were provided by the Korea Economy Daily and Acrofan.

The authors thank Cheoneum Park for discussions about task selection and DP task, Jinhyuk Lee and Minjoon Seo for discussions on MRC task, Sujeong Kim and DongYeon Kim for considerable efforts to manage the annotation for MRC dataset, and Sangah Park for careful consideration of data construction in DP, NER, and RE. We thank Junyeop Lee, Geonhee Lee, Jiho Lee, Daehyun Nam, and Yongjin Cho for the leaderboard and the evaluation system.

This study is reviewed and approved by the KAIST Institutional Review Board (#KH2020-173).

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
