# OpenReview forum: "KLUE: Korean Language Understanding Evaluation"
_NeurIPS.cc/2021/Track/Datasets_and_Benchmarks/Round2 — NeurIPS 2021 Datasets and Benchmarks Track (Round 2)_

### Official Review · Reviewer_kbEC · 2021-09-14
**A Korean NLU benchmark**

**Rating:** 6
**Confidence:** 4
**Correctness:** Yes. But a limitation is that no huma…
**Clarity:** Yes.

**Strengths:**

1. The first general NLU benchmark for Korean.
2. All tasks introduced in KLUE are newly annotated with careful considerations.
3. Several new Korean pre-trained language models are introduced and evaluated on KLUE.

**Weaknesses:**

1. Though I acknowledge the huge works by the authors, including constructing every task and pre-trained models, etc., the novelty seems to be limited, as most of the annotation procedures are not new. I'm not sure whether a non-Korean researcher could learn something new from this paper, other than its solid contribution to the Korean NLP community.
2. Another concern is that the proposed benchmark might only be interested in a limited research community (for studying Korean NLP). And I am not sure if it is suitable for NeurIPS.
3. It is not clear whether the data for pre-training PLMs will also be released.
4. There is no human performance (or estimated) on KLUE.

**Additional Feedback:**

1. Line 136: superGLUE -> SuperGLUE
2. Possible missing reference for the whole word masking strategy (line 258). And also, what tool was used for Korean word segmentation for WWM?
3. Unlike most of other NLU benchmarks, there is no overall score in KLUE. It is quite hard to compare different PLMs as a whole. If this is not designed, I'd suggest adding an overall score to your benchmark leaderboard.
4. line 292: The illustration 'a few interesting observation' is somewhat confusing, as these observations are quite normal and expected.
5. Will the pre-training model and data also be released to the public?

**Documentation:**

Documentation of this paper is thorough.

**Ethics:**

None.

**Relation To Prior Work:**

Well-discussed.

**Summary And Contributions:**

This paper proposes Korean Language Understanding Evaluation (KLUE), consisting of 8 NLU tasks, including classification, regression, machine reading comprehension, NER, etc. Different from other NLU benchmarks, all eight tasks in KLUE are newly annotated. To examine the benchmark, several new Korean pre-trained language models are built, which are based on BERT/RoBERTa. The results show that the KLUE-RoBERTa_large could achieve the best performance on most tasks than mBERT or XLM-R.

---

> ### Author Response · Authors · 2021-09-29
> **Response to Reviewer kbEC**
>
> Thank you for acknowledging our contributions.
>
> (Little novelty and impact) Our main contribution is building a Korean NLU benchmark suite from scratch while following the 4 design principles. Each task has pushed forward the annotation quality addressing the limitations of previous datasets. Our procedures and decisions can give insights to future works on multi-/cross-linguality (e.g., XTREME) or similarly-low-resourced languages, which is not limited to the Korean NLP community or researchers. Another major contribution is releasing the pre-trained models for Korean. With the models, even if some results were predictable, we present simple yet effective baseline models and their performances in downstream tasks. We also present the effect of tokenization and pseudonymization of pretraning corpora in Appendix L.
>
>
> (Releasing pre-training corpus and pretrained model) We publicly released our pretrained models at https://huggingface.co/klue. On the other hand, we did not open the pretraining corpora, which are all created from publicly available texts, in order to avoid any issues including intellectual property in the future, which is in contrast to KLUE. We elaborated more in Appendix M.
>
> (Human Performance) We measured human performance on KLUE-MRC whose construction process - finding answer span and then generating question - is opposed to the task-solving process - reading a question and then finding an answer span - as described in Appendix I.3. We agree that measuring human performance is important, but for the other tasks, we were limited in time and budget. We will pursue this direction in future work.
>
> (Overall scoring) For the overall scoring of an NLU model, we avoid conventional averaging of all scores gained from each task in KLUE as we mentioned in the paper. The performance of all tasks is measured by different evaluation metrics. This is because we carefully choose the metric for each task while considering its own characteristics. Their granularity differs by tasks, for example, KLUE-MRC and KLUE-NER employ character-level metrics because an entity can exist within a word in Korean whereas KLUE-STS and KLUE-NLI use sentence-level metrics. Furthermore, we use various metrics across tasks, such as F1 score, accuracy, area under the curve, UAS, LAS, ROUGE-W, joint goal accuracy, and Pearson’s correlation. In this situation, simply averaging all task scores results in misleading overall performance evaluation. The average will lose its interpretability as well as giving higher weights to a certain task in unintended ways.
> To the best of our knowledge, analyzing the correctness of a model’s prediction by using Item Response Theory (IRT) framework to estimate such capability is proposed recently, however, we ﬁnd that it is not clear how it should be applied precisely in our benchmark. If there is a suitable alternative for our situation, we are happy to apply it as an overall scoring method. These are included in Appendix N, and we will clarify the answer in the main page.

---

### Official Review · Reviewer_isyR · 2021-09-20
**Nice work!**

**Rating:** 8
**Confidence:** 4
**Clarity:** Yes, this paper is well written with …

**Strengths:**

The paper is well written with a well-designed structure. The three contributions listed above are significant, especially the third one sets up a typical example for future similar studies. The accessibility and accountability are clear with the data source and nice documentation.

**Weaknesses:**

I have not found any important shortcomings in this paper. To make the paper stronger I would suggest extending the part of design principles to clarify the relationships to each task. This will give more insights for future studies.

**Additional Feedback:**

I look forward to seeing this paper accepted.

**Correctness:**

Yes, the dataset is constructed in a sound way and the released pre-trained language models are reliable.

**Documentation:**

Yes, this paper provides an outstanding example for future works of similar aims.

**Ethics:**

No.

**Relation To Prior Work:**

Yes, the discussion of related work is detailed.

**Summary And Contributions:**

This paper ***1) introduces Korean Language Understanding Evaluation (KLUE) benchmark***, including eight different natural language understanding (NLU) tasks. Meanwhile, the authors ***2) release pre-trained language models*** and ***3) give insights for future works building similar resources***, both of which are equally valuable for the community.

---

> ### Author Response · Authors · 2021-09-29
> **Response to Reviewer isyR**
>
> Thank you for suggesting that we add the design principles to give even more insights to future studies. Below, we discuss the relationship between design principles and each task in KLUE. These are our high-level considerations for all 8 tasks, and if you find these helpful, we will add them to the paper.
>
> Design Principle 1: Covering diverse tasks and corpora:
> - In GLUE, for example, there are 4 natural language inference datasets (MNLI, RTE, QNLI, WNLI) and 3 similarity and paraphrase datasets (MRPC, QQP, STS-B). We take the approach of diversifying the tasks, including only one dataset for each task. We diversify the task format as well, including sentence classification (YNAT), sentence pair regression (KLUE-STS), sentence pair classification (KLUE-NLI), character-level classification (KLUE-NER), word-level classification (KLUE-DP), sentence classification with entities (KLUE-RE), span prediction (KLUE-MRC), slot-value prediction (WoS). Lastly, we collect 10 corpora to incorporate diverse domains and styles and use them across various tasks. Details of the source corpora assignment are in Appendix B.
>
> Design Principle 2: Accessible to everyone without any restriction:
> - We ensure any corpora and resources in KLUE can be freely copied, redistributed, remixed, and transformed for the purpose of benchmarking NLU systems. This was possible because we carefully chose source corpora in terms of the license, which allows free access after annotation.
>
> Design Principle 3: Obtaining accurate and unambiguous annotations:
> - We carefully design annotation guidelines of all tasks and improve them over multiple iterations, to assure accurate annotations.
>
> Design Principle 4: Mitigating ethical issues in PLMs:
> - We remove examples from both unlabeled and labeled corpora that reflect social biases, contain toxic content or personally identifiable information (PII), both manually and automatically. While acknowledging and mitigating bias and toxic content is important for corpora used for training LMs, smaller and more focused benchmark datasets would not benefit from biased or toxic content.

---

### Official Review · Reviewer_EQss · 2021-09-20
**A new language understanding benchmark in Korean**

**Rating:** 7
**Confidence:** 4
**Correctness:** seems good

**Strengths:**

- Large scale dataset on wide variety of language understanding task
- A unified benchmark for evaluating Korean (13th most used language)  language models in the world will encourage research in Korean language.
- Benchmark has been built from ground up instead of translating existing English benchmark into Korean.

**Weaknesses:**

An evaluation with existing English benchmarks like GLUE translated to Korean language would have served as one weak baseline.

**Additional Feedback:**

Read abstract and check it for Grammatical mistakes and correct it.

**Clarity:**

Overall the paper is well written. But paper could have been structured in a bit better manner. Also, papers need one careful reading for grammatical mistakes.


**Documentation:**

Yes, data collection and other details are clearly explained in the paper.

**Relation To Prior Work:**

Probably limitation of existing English dataset translated to Korean language could have been discussed in a bit detailed manner.

**Summary And Contributions:**

Authors build a new language understanding benchmark for Korean language. They build dataset for 8 NLU tasks namely topic classification, semantic textual similarity, natural language inference, named entity recognition, relation extraction, dependency parsing, machine reading comprehension and dialogue state tracking. Along with the dataset authors also release pre-trained model for Korean language based on BERT, RoBERTa etc. Authors have documented details about benchmark construction and testing process.

---

> ### Author Response · Authors · 2021-09-29
> **Response to Reviewer EQss**
>
> Thank you for supporting our effort to foster more research in Korean.
>
> (Translating English benchmark to Korean) Translating existing English benchmarks to other languages is widely used in building cross-lingual/low-resource language benchmarks, so translation could be employed to build Korean benchmarks as well. Thus we agree that it is meaningful to compare translated GLUE with KLUE to investigate whether the translation can be an effective way to build reliable Korean NLU benchmarks.
>
> We tried this with NLI, which is one of the more straightforward tasks for comparison. This is presented in the Appendix, where we compare the quality of the dev/test set in KorNLI (Human-translated examples of XNLI. They were the first machine translated, and then post-edited by professionals) against KLUE-NLI by re-annotating the label. In 30% of KorNLI examples, human annotators failed to recover the gold labels, but for KLUE-NLI they perfectly recovered the gold labels. This direct comparison is not possible for all tasks, and the discrepancy may differ by task, but we expect that because of the dissimilarity of the two languages, the KLUE benchmark will be significantly more appropriate than translated GLUE.
>
> (Grammatical Issues) We will carefully revise the manuscript to improve the grammar and other writing details.

---

### Decision · Program_Chairs · 2021-10-09

**Decision:**

Accept

**Comment:**

This work presents KLUE, which provides a set of benchmarks for natural language understanding. All reviewers agree this is high-quality work, and that the process of creating data was done in a thoughtful manner, with attention to detail. High-quality datasets in a multitude of datasets are indispensable and this work will contribute to work on low resource languages in general.